# The Role of the Respiratory Microbiome and Viral Presence in Lower Respiratory Tract Infection Severity in the First Five Years of Life

**DOI:** 10.3390/microorganisms9071446

**Published:** 2021-07-05

**Authors:** Ivo Hoefnagels, Josephine van de Maat, Jeroen J.A. van Kampen, Annemarie van Rossum, Charlie Obihara, Gerdien A. Tramper-Stranders, Astrid P. Heikema, Willem de Koning, Anne-Marie van Wermerskerken, Deborah Horst-Kreft, Gertjan J.A. Driessen, Janine Punt, Frank J. Smit, Andrew Stubbs, Jeroen G. Noordzij, John P. Hays, Rianne Oostenbrink

**Affiliations:** 1Department of General Pediatrics, Erasmus MC–Sophia Children’s Hospital, 3015GD Rotterdam, The Netherlands; i.hoefnagels@erasmusmc.nl (I.H.); Josephine.vandeMaat@radboudumc.nl (J.v.d.M.); 2Department of Internal Medicine, Radboud Center of Infectious Diseases, Radboudumc, 6525GA Nijmegen, The Netherlands; 3Department of Viroscience, Erasmus University Medical Centre (Erasmus MC), 3015GD Rotterdam, The Netherlands; j.vankampen@erasmusmc.nl; 4Departement of Pediatrics, division of Pediatric Infectious Diseases, Erasmus MC–Sophia Children’s Hospital, 3015GD Rotterdam, The Netherlands; a.vanrossum@erasmusmc.nl; 5Department of Pediatrics, Elisabeth-TweeSteden Hospital, 5042AD Tilburg, The Netherlands; c.obihara@etz.nl; 6Department of Pediatrics, Franciscus Gasthuis & Vlietland, 3045PM Rotterdam, The Netherlands; g.tramper@franciscus.nl; 7Department of Medical Microbiology and Infectious Diseases, Erasmus University Medical Centre (Erasmus MC), 3015GD Rotterdam, The Netherlands; a.heikema@erasmusmc.nl (A.P.H.); d.kreft@erasmusmc.nl (D.H.-K.); j.hays@erasmusmc.nl (J.P.H.); 8Department of Pathology, Clinical Bioinformatics Unit, Erasmus University Medical Centre (Erasmus MC), 3015GD Rotterdam, The Netherlands; w.dekoning.1@erasmusmc.nl (W.d.K.); a.stubbs@erasmusmc.nl (A.S.); 9Department of Pediatrics, Flevoziekenhuis, 1315RA Almere, The Netherlands; avwermeskerken@flevoziekenhuis.nl; 10Department of Pediatrics, Maastricht University Medical Center, 3584CX Maastricht, The Netherlands; gertjan.driessen@mumc.nl; 11Department of Pediatrics, Langeland Ziekenhuis, 2725NA Zoetermeer, The Netherlands; j.punt@llz.nl; 12Department of Pediatrics, Maasstad Ziekenhuis, 3079DZ Rotterdam, The Netherlands; smitf@maasstadziekenhuis.nl; 13Department of Pediatrics, Reinier de Graaf Gasthuis, 2625AD Delft, The Netherlands; j.noordzij@rdgg.nl

**Keywords:** lower respiratory tract infection, respiratory microbiome, virus, nanopore sequencing, 16S-rRNA gene

## Abstract

Lower respiratory tract infections (LRTIs) in children are common and, although often mild, a major cause of mortality and hospitalization. Recently, the respiratory microbiome has been associated with both susceptibility and severity of LRTI. In this current study, we combined respiratory microbiome, viral, and clinical data to find associations with the severity of LRTI. Nasopharyngeal aspirates of children aged one month to five years included in the STRAP study (Study to Reduce Antibiotic prescription in childhood Pneumonia), who presented at the emergency department (ED) with fever and cough or dyspnea, were sequenced with nanopore 16S-rRNA gene sequencing and subsequently analyzed with hierarchical clustering to identify respiratory microbiome profiles. Samples were also tested using a panel of 15 respiratory viruses and *Mycoplasma pneumoniae*, which were analyzed in two groups, according to their reported virulence. The primary outcome was hospitalization, as measure of disease severity. Nasopharyngeal samples were isolated from a total of 167 children. After quality filtering, microbiome results were available for 54 children and virology panels for 158 children. Six distinct genus-dominant microbiome profiles were identified, with *Haemophilus*-, *Moraxella*-, and *Streptococcus*-dominant profiles being the most prevalent. However, these profiles were not found to be significantly associated with hospitalization. At least one virus was detected in 139 (88%) children, of whom 32.4% had co-infections with multiple viruses. Viral co-infections were common for adenovirus, bocavirus, and enterovirus, and uncommon for human metapneumovirus (hMPV) and influenza A virus. The detection of enteroviruses was negatively associated with hospitalization. Virulence groups were not significantly associated with hospitalization. Our data underlines high detection rates and co-infection of viruses in children with respiratory symptoms and confirms the predominant presence of *Haemophilus*-, *Streptococcus*-, and *Moraxella*-dominant profiles in a symptomatic pediatric population at the ED. However, we could not assess significant associations between microbiome profiles and disease severity measures.

## 1. Introduction

Lower respiratory tract infections (LRTIs) are the leading global cause of mortality in children under 5 years old [1], with 13% of all deaths attributed to LRTIs in 2016 [2]. In addition, LRTIs are a major cause of morbidity, as 53–62% of hospitalizations for infectious diseases were attributed to LRTI [3]. Nonetheless, the majority of LRTI cases are mild. Therefore, the ability to adequately differentiate between mild and severe LRTI cases could be important for guiding the treatment strategy and improving LRTI outcomes.

LRTIs in children are most commonly caused by viruses [4], with respiratory syncytial virus (RSV) and influenza virus being most often implicated as causative pathogens [5]. RSV and human metapneumovirus infections have been reported to pose the greatest risk for hospitalization [6]. Bacterial LRTIs in children are commonly associated with *Streptococcus pneumoniae* and *Haemophilus influenzae* infections [5].

An important role for respiratory health has been attributed to the respiratory microbiome, with reported associations between respiratory microbiome profiles and respiratory related health and disease. The respiratory microbiome also plays a role in immune system development and influences immune responses, as do local viral and bacterial interactions. [7,8]. Recent reports have demonstrated associations between the upper respiratory tract microbiome and the susceptibility and severity of LRTIs in asthmatic, health versus disease, and bronchiolitic cohorts of children [9,10,11,12,13]. By combining viral, respiratory microbiome, and host-related data, it has been reported that children with LRTIs can be differentiated from healthy controls [11]. Additionally, specific microbiome clusters have been associated with specific viral LRTIs and disease severity. For example, *Haemophilus influenzae*-enriched and *Streptococcus*-enriched clusters have been described to be positively associated with RSV infection and RSV-related hospitalization [13]. Yet to be discovered is whether these insights can contribute to prediction of a severe disease course within children presenting at the emergency department (ED) with symptoms of LRTIs.

In this current study, we aimed to describe the viral and microbiome spectrum and assess the potential contribution of clinical parameters, and viral infection and respiratory microbiome data, in helping to predict potential severe disease in children aged 1 month to 5 years presenting at the ED with suspected LRTIs. This study could improve our understanding of the implications of the interaction between respiratory microbiome and LRTIs and contribute to future clinical disease severity differentiation with potential implementation in clinical decision making in EDs.

## 2. Materials and Methods

### 2.1. Population and Study Design

The study population originated from a previously described cohort of 999 children aged one month to five years [14]. Briefly, children were included in the study between 1 January 2016 and 30 September 2018 by treating physicians, when they presented at the ED with fever (temperature > 38.5 degrees Celsius) and cough or dyspnea as symptoms of a potential RTI. The eight participating centers included one tertiary (Erasmus MC Rotterdam) and seven general hospitals (Maasstad Hospital Rotterdam, Franciscus Vlietland Hospital Rotterdam, LangeLand Hospital Zoetermeer, Reinier de Graaf Hospital Delft, HAGA Hospital The Hague, Elisabeth-Tweesteden Hospital Tilburg, Flevo Hospital Almere) all located in the Netherlands. Exclusion criteria were defined by an increased risk of a complicated disease course, reported in more detail previously [14]. In short, an increased risk of complicated disease course included children with relevant comorbidities (immunodeficiency, congenital heart defects, chronic pulmonary disease, multiple handicaps, or prematurity), signs of complicated pneumonia, another infectious focus, antibiotic use within a week prior to the ED visit, or an amoxicillin allergy. In our study, we solely included children from whom a nasopharyngeal sample was obtained for microbiome and/or virology assessment.

### 2.2. Outcomes

The primary outcome was the rate of hospitalization as a measure of disease severity. Secondary outcomes were strategy failure and the predicted risk of having a bacterial LRTI. Strategy failure was a predefined composite outcome, based on disease course during the first 7 days after ED presentation. Predicted risk of bacterial pneumonia was computed with the Feverkidstool, a previously validated model to predict the presence of pneumonia or other serious bacterial infections in children with fever [14,15,16]. Three separate risk groups were defined, with cut-off points set at <3% for low predicted risk for bacterial pneumonia, 4–10% for intermediate and >10% for high predicted risk [14]. Detailed definitions of outcome measures are available in the online supplement.

### 2.3. Data and Sample Collection

Clinical parameters assessment and CRP (C-reactive protein) point of care (bedside) testing at ED presentation were performed by the attending nurse and physician. Extensive general and clinical data were collected on an electronic case record form. Follow-up data 7 days post initial presentation were collected through structured telephone calls or collected directly from patients and parents during the period of hospitalization. Nasopharyngeal aspirates (NFA) were obtained by the attending nurse of physician during the first ED visit, using 0.9% NaCl, 1 mL in each of the nostrils. One sample was obtained per individual. Samples were stored at −80 degrees Celsius.

### 2.4. Microbiome Sample Processing

Analysis of the microbiome was performed on an extract of the NFA during a 2 day period and two negative controls (assay buffer only) were included in the extractions to check for potential contamination. 16S-rRNA sequencing was performed using a MinIon nanopore sequencer (Oxford Nanopore Technologies (ONT), Oxford, UK). The microbiome methodology, including complete DNA extraction and sequencing protocol, was verified in a parallel study (presented in a separate manuscript [17]) using defined single nasal microbiota species (obtained from the American Type Culture Collection, ATCC, Manassas, VA, USA) and is presented in detail in the online supplement. In short, DNA extraction was performed using the AGOWA mag minikit (NAP40402, LGC Genomics, Berlin, Germany), and manufacturer’s instructions were followed. Nasal swab fluid with lysis buffer added to lysing matrix was compared to lysis buffer as negative controls. DNA concentrations were standardized using the Quant-iT PicoGreen dsDNA Assay kit (P7589, Invitrogen, Carlsbad, CA, USA), followed by 16S rRNA gene sequence library preparation. Basecalling of sequence reads was performed using Guppy software (version 3.1.5+781ed57, ONT, Oxford, UK). Debarcoding and classification of bacteria was performed using the EPI2ME (version 3.2.2, ONT, Oxford, UK) 16S workflow. Finally, a quality filtering step was performed was performed using a custom script with an average Q score set to 9 and identity of 85%.

Sequenced samples were initially filtered for a minimum number of 1000 reads per sample, as has been standardized in other nasal microbiome studies based on Illumina sequencing [18,19,20]. Rarefaction curves of reads versus richness were constructed to determine the minimum number of reads required per sample to detect all bacterial species. Further quality filtering (read minimum, removal of rare taxa with less than 0.05% relative abundance across all samples) was performed in Calypso (University of Queensland, Brisbane, Australia) [21].

### 2.5. Virology Procedures

Samples were analyzed using real-time reverse transcriptase PCR for detection of fifteen respiratory viruses and *Mycoplasma pneumoniae*. The viruses tested were adenovirus (ADV), human bocavirus (HBoV), enterovirus, human metapneumovirus (hMPV), human rhinovirus (HRV), influenza A virus and influenza B virus, parainfluenza virus (PIV) types 1 through 4, respiratory syncytial virus (RSV) type A and B, and human coronavirus types NL63, OC43, and 229E. Samples were considered positive for a certain virus when cycle threshold (Ct) values were <40. Subtypes of viruses were clustered together for further analyses and reporting, except for influenza A virus and influenza B virus. For the real-time reverse transcriptase PCR procedures, we followed the laboratory procedures as reported elsewhere [22,23].

Patients were categorized according to the virulence of the viruses detected in their sample, where virulence was the previously reported association of the virus with symptoms in children relative to asymptomatic controls. High virulence was assigned to those positive for hMPV, influenza type A and B, parainfluenza virus and RSV, and low virulence to children positive for bocavirus, adenovirus, coronavirus, enterovirus, and rhinovirus [24,25,26,27,28,29,30,31]. Patients with multiple viruses were assigned to the high virulence group if positive for any of the high virulence viruses. High and low virulence groups were used for further analyses.

### 2.6. Analysis

#### 2.6.1. Microbiome Data Analysis

Count data were normalized using total sum scaling (TSS), where read counts are divided by the total number of reads in each sample. Relative abundance of the top 20 genera and families across all samples were calculated. Unsupervised hierarchical clustering with the Bray–Curtis distance metric was performed to identify clusters. For each identified cluster, the most abundant taxon at the genus level within the cluster was used as a classifier taxon and clusters were named accordingly. Clusters were checked for plausibility and distinctiveness by bar charts containing the top 20 genera and non-metric multidimensional scaling plots. Richness (number of taxa) and diversity (Shannon index) are presented in bar charts and differences were tested by the Kruskal–Wallis test. Microbiome data analyses were performed using Calypso Version 8.84 (University of Queensland, Brisbane, Australia) [21].

#### 2.6.2. Statistical Analysis

For the outcome measures hospitalization, strategy failure, and predicted risk of having a bacterial LRTI, we preformed univariate analyses across viruses detected in ≥10 cases, across virulence groups, and across identified microbiome profiles. We used either chi-squared test or Fishers’ exact test for categorical outcomes and, for numerical outcomes, one way-ANOVA for normal distributed variables or Kruskal–Wallis test as a non-parametric test. Identified differences were considered significant if *p* < 0.05.

Univariate unadjusted associations between microbiome profiles, virology data and clinical parameters, and hospitalization and strategy failure were explored. Those significantly associated with the outcome measure were included in the adjusted multivariate logistic regression model. Predictors of interest were adjusted for age, gender, presence of any virus, and calculated risk of bacterial LRTI. For multivariate models, we assumed any missing data to be missing at random and handled missing data by performing multiple imputation (m = 10) including all variables included in the analysis in the imputation model. All statistical analyses were performed using IBM SPSS version 25 (IBM Corp., Armonk, NY, USA).

## 3. Results

### 3.1. Study Population

Of the 999 children included in the original STRAP cohort, data for microbiome profiling, virology, or both were available for 167 children. Ages ranged between 1 and 54 months, with a median age of 17 months; 62.3% were male. Children presented with a median temperature of 38.8 °C and the most common presenting symptoms were cough (94.5%), followed by dyspnea (79.6%) and rales (56.2%). Additional baseline demographic, clinical characteristics, and outcome variables are presented in Table 1 and Appendix A. A microbiome profile was available for 54 children after quality filtering in Calypso [21]. The 1000 read cut-off as pre-planned and described in the method section was altered to a 10,000 read cut-off based on the results of a rarefaction curve graph. For virology analyses, 158 reliable sample results were available. A flowchart of sample inclusion and processing is shown in Appendix A.

### 3.2. Virology

Of 158 samples, 139 (88.0%) were positive for at least one pathogen, with 94 (67.6%) samples testing positive for a single virus, 32 samples (23.0%) for two viruses, and 13 samples (9.4%) for more than two viruses. The most common viruses were rhinovirus (n = 63, 39.9%) and RSV (n = 39, 24.7%). The least frequently found were influenza A virus (n = 5) and influenza B virus (n = 2) (see Table 2).

The composition of viral coinfections varied greatly among the viruses tested. Adenovirus, bocavirus, and coronavirus were found together with at least one other virus in 90.5%, 90.9%, and 80% of cases, respectively. For hMPV, this was true for only 25% of cases, and influenza A virus was found only as a single virus infection (see Table 2). Children with viral co-infections were significantly younger than children with a single virus or virus negative samples (11 vs. 19.5 vs. 21 months, *p* = 0.002).

### 3.3. Association between Viral Infection and Clinical Outcome Measures

Univariate analyses did not show significant differences between virulence groups for any of the outcome measures, although we did see a trend towards more hospitalization, strategy failure, and higher predicted risk of bacterial pneumonia for the high virulence group (Appendix A).

Further separate unadjusted analyses for individual viruses showed that enterovirus was significantly associated with lower rate of hospitalization (OR (odds ratio) 0.076, 95% CI 0.016–0.353, *p* = 0.001). A similar tendency was observed for adenovirus (OR 0.46, 95% CI 0.182–1.161, *p* = 0.1). In the multivariate analysis, this difference remained significant for enterovirus (OR 0.093, 95% CI (confidence interval) 0.019–0.464, *p* = 0.004) and not for adenovirus (OR 0.388, 95% CI 0.137–1.090, *p* = 0.074), after adjusting for viral infection, age, gender, and Feverkidstool risk groups (see Table 3). Multivariate analyses were not performed for associations between individual viruses and strategy failure because the *p*-value for each virus was >0.15 in unadjusted analyses.

### 3.4. Microbiome

Relative abundance of microbiota read counts at genus and family levels are presented in Appendix A. Given the explorative nature of this study, we used several analyses to define microbiome profiles. First, hierarchical clustering revealed six clusters that were obtained using the visual representation of the Bray–Curtis distance presented in the dendrogram and heatmap in Figure 1. The clusters were named as genus dominant microbiome profiles, based on the most abundant taxa on genus level within the cluster, as visualized in Figure 2. The *Haemophilus*-dominant profile was the most common (42.7%), followed by the *Moraxella*-dominant profile (38.9%) and the *Streptococcus*-dominant profile (13.0%). Three samples were deemed outliers because they were distinctly dissimilar to any of the other samples (see Figure 1). A non-metric multi-dimensional scaling (NMDS) plot, in which the previously mentioned profiles are visualized, showed distinct multidimensional distances between the different genus-dominant profiles (Appendix A). The relative abundance of the 20 top genera in each sample is presented in Figure 2. A single genus or family dominated most samples. However, within the *Moraxella*-dominant profile, there was relatively high abundance of *Carnobacteriaceae*. At sequence read level (reads that could be assigned to species level), both richness and diversity (Shannon index) were significantly lower in *Haemophilus* compared to *Moraxella*-dominant profiles (Appendix A).

Univariate analyses showed no significant differences between microbiome profiles for any of the clinical outcome measures (Appendix A). In both unadjusted and adjusted analysis, microbiome profiles were not significantly associated with the rate of hospitalization (see Table 3). Modelling was not performed for strategy failure because there were <10 strategy failure cases within children with a microbiome profile present.

### 3.5. Integrating Viral, Microbiome, and Clinical Data

Clinical descriptives for the observed microbiome profiles are presented in Appendix A. Virulence groups did not differ significantly among the microbiome profiles (*p* = 0.697). Among the microbiome profiles, we observed no differences in viral presence for those viruses present in more than 10 cases (i.e., hMPV, RSV, and rhinovirus). In the absence of significant associations between microbiome profiles and severity measures we did not further explore models for hospitalization or strategy failure including clinical, virology, and microbiome predictors.

## 4. Discussion

The current publication describes a follow-up study of the STRAP trial with children aged one month to five years, who presented at the ED with fever and symptoms of RTI. We detected high numbers of viruses and viral co-infections in their respiratory samples. In multivariate analyses, enterovirus was negatively associated with hospitalization. We identified three major microbiome profiles, dominated by either *Haemophilus*, *Moraxella*, or *Streptococcus*. The presence of viruses was similar among the microbiome profiles. We did not observe significant associations between microbiome profiles and hospitalization, strategy failure, or predicted risk of bacterial pneumonia.

Our high viral detection rate (88%) was comparable to that of other studies with similar populations, which reported detection rates ranging from 61% to 97% [6,11,25,27,29,32,33]. Interestingly, viral detection in healthy populations was also reported to be high, with reported viral detection rates in healthy children ranging from 24.4% to 83% [11,25,27,29], questioning the causal relationship of detected viruses with respiratory symptoms. Differences in detection rate and proportion of co-infections between studies could be related to the difference in viral panels tested, with our comprehensive panel of 15 viruses contributing to our high detection rate. Additionally, differences in age distributions of study participants might have influenced detection rates [34]. In our study, children with multiple viruses were significantly younger than children with a single virus detected, in contrast with earlier studies that reported children with coinfections to be significantly older [35,36,37]. As in previous studies in symptomatic children (including those with severe LRTIs), RSV and rhinovirus were the most common [6,38,39]. In this study, the proportion of co-infections was high, but consistent with other studies, which reported co-infection rates ranging from 18% to 41% [6,32,36,40,41]. Bocavirus and adenovirus were frequently found together with other viruses, which is confirmed by other reports [6,32,36,40,41]. With respect to differentiation of virus results into virulence groups, we did not see significant differences among viral virulence groups and study outcomes, although we did see a trend towards more hospitalization, strategy failure, and higher predicted risk of bacterial pneumonia for the high virulence group.

The six distinct identified microbiome profiles we identified have considerable overlap with the identified profiles in other studies. In particular, *Haemophilus*-, *Moraxella*-, and *Streptococcus*-dominant profiles are identified frequently [9,12,13,42,43]. However, we did not identify the *Corynebacterium*-dominant profile in our ED study population, in contrast to studies focusing predominantly on healthy children [9,13,42]. A possible explanation for this is that *Corynebacterium* are considered to be protective [42,44,45]. Additionally, *Corynebacterium*- and *Staphylococcus*-dominated dominated profiles have mainly been observed in younger children up to two years of age [42,44], whereas we included children up to five years of age. Also of note is that a recent publication by Heikema et al. [17], indicated possible problems with the detection of *Corynebacteria* using nanopore-based amplification primers compared to Illumina-based primers. Further, with respect to nanopore sequencing, we found a relatively large number of sequencing reads that could not be classified to genus level. This may have been a consequence of sequencing errors introduced by the nanopore flowcell that was used for this study (R9.2). However, nanopore sequencing is constantly evolving, with more recent evidence indicating a continued advancement in the ability of nanopore flowcells to identify the nasal microbiome at the genus level [17].

Previous studies have reported inconsistent associations between distinct nasopharyngeal microbiome profiles and disease severity measures [12,13,42]. Although they all identified *Haemophilus*-, *Moraxella*-, and *Streptococcus*-dominant profiles, among other profiles, a relative abundance of *Streptococcus* was associated with influenza and hospitalization [13], intensive care admission was lowest in *Moraxella*-dominant profiles and highest in *Haemophilus*-dominant profiles [12], and *Streptococcus*-, *Haemophilus*-, and *Moraxella*-dominant profiles were significantly associated with respiratory symptoms, with the *Moraxella*-dominant profile associated with the severity of RSV infections [42]. We did not find significant differences between the three profiles and the disease severity outcome measures. This may result from our rather homogeneous population of symptomatic children visiting the ED, with a high hospitalization rate. However, in this ill population, we did confirm the predominant presence of the *Haemophilus*-, *Streptococcus*-, and *Moraxella*-dominant profiles.

### Strengths and Limitations

Our study is one of the first to combine comprehensive high quality clinical data with extensive viral detection and nanopore 16S-rRNA sequencing of the respiratory microbiome.

Although we focused on LRTIs, samples were collected from the nasopharynx. Some studies indicate that the respiratory tract microbiome in general is better represented by other samples, such as tracheal aspirates, sputum, or bronchial aspirate lavages [46]. When comparing sick versus healthy children, however, it has been reported that the nasopharynx is an appropriate proxy for the lower respiratory tract microbiome in LRTIs [11]. From our observations, the use of upper respiratory tract microbiome profiling to differentiate disease severity within a cohort of children who are all presenting with symptoms of a LRTI may not yield statistically significant results. As our study was relatively small, further research is required in this area because the use of microbiome profiling of the upper respiratory tract to identify LRTIs has several advantages in terms of invasiveness and clinical and ethical viability compared to lower respiratory tract microbiome sampling.

Because collecting microbiome and virology data was a secondary aim of the original STRAP trial, the study was not designed specifically, nor powered accordingly, for the present analyses. Sample collection and analyses was therefore not implemented equally across participating hospitals and sample numbers were relatively low. This could lead to selection bias. However, our population closely resembles that of the original study population of the STRAP trial in terms of baseline characteristics, including age and gender [14]. Additionally, the data obtained regarding viral (co-)infections and microbiome profiles were similar to those of previous published reports, as described above.

With respect to virological analysis, we clustered virus subtypes into a single type (i.e., RSV, coronavirus, and parainfluenza virus) to increase the statistical validity of the results. However, this could have led to underestimation of potentially harmful subtypes of certain viruses. For RSV, equal severity of type A and B infection was assumed based on reported resemblance [47,48].

Most studies on respiratory microbiome in children have used Illumina sequencing, whereas we worked with nanopore sequencing. Nanopore sequencing has several potential advantages, including long-read output, real-time analyses, and portability, but has relatively lower accuracy compared to short-read methods [49,50]. The long-read output, which in microbiome research could be used to sequence the full 16S-rRNA gene, could potentially lead to more accurate assembly of genome data, especially at the species level. However, the error rate in basecalling of nanopore sequencing averages up to 10% [50], meaning that microbiome profiling at the level of species, and sometimes even genus, may be problematic, as observed in this study. However, as previously mentioned, advances in nanopore sequencing technology occur regularly and newer flowcell sequencing cartridges appear to have reduced sequencing error rates [17]. One consequence of the potentially relatively high error rate encountered in this study was observed during rarefaction analysis, i.e., a mathematical calculation plotting the number of species against the number of samples (to generate a graph showing the number of sequencing reads against microbiota richness). Ideally, for the sample size used, a rarefaction analysis plot should show a steadily decreasing slope and plateau, with the plateau indicating that the number of samples is sufficient to obtain data from all species present within a particular microbiome. Previous publications using Illumina-based sequencing of the upper respiratory microbiome have indicated a rarefaction plateau at a minimum of approximately 1000 reads. However, from our nanopore results, a plateau was not observed, even for a sample that generated 100,000 reads. The most likely explanation for this discrepancy is the presence of errors in the nanopore sequence results that we obtained. Therefore, as a compromise, we only used samples containing a minimum of 10,000 reads for analysis and filtered out rare taxa with less than 0.05% relative abundance, maximizing the balance between accuracy and the number of remaining samples for analyses. Subsequent analysis of our microbiome data showed that the microbiome profiles obtained in this study resembled previously reported upper respiratory tract microbiome profiles obtained from symptomatic children using Illumina sequencing. Finally, in general, the range of clustering techniques used in different studies, such as partitioning around medoids (PAM) clustering [10,12,51] and several hierarchal clustering techniques [13,42] may impact the result comparability between microbiome-based studies.

## 5. Conclusions

This study underlines the high detection rate of viruses and viral co-infections in children with respiratory symptoms. It also highlights the presence of distinct microbiome profiles in the nasopharyngeal tract of children with respiratory symptoms at the ED, with *Haemophilus*-, *Moraxella*-, and *Streptococcus*-dominant profiles being the most prevalent. Although we could not assess significant associations between microbiome profiles and disease severity measures (hospitalization, strategy failure, and predicted risk of bacterial pneumonia), the predominant detection of the aforementioned profiles, compared to profiles often found in healthy children, indicates potential associations between microbiome composition and LRTIs. Further sufficiently powered research should address these relationships, and could progress to identifying and potentially implementing microbiome profiling to be used as a biomarker to identify children at risk of severe LRTIs, in combination with existing clinical severity biomarkers in EDs.

## Figures and Tables

**Figure 1 microorganisms-09-01446-f001:**
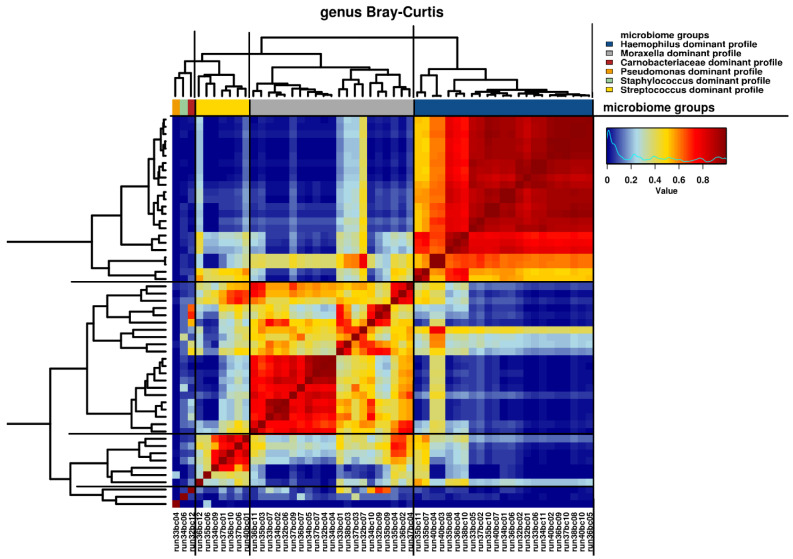
Hierarchical clustering of microbiome samples. Visual representation of hierarchical clustering of microbiome samples at the genus level with the Bray–Curtis distance metric. A heat map is shown, characterized by similarity values ranging from 0 to 1. Samples are presented as run numbers. A dendrogram of the hierarchical clustering is shown with distances represented by the length of the individual branches. Genus dominant profiles are presented by colored bars, as indicated by the accompanying figure legend.

**Figure 2 microorganisms-09-01446-f002:**
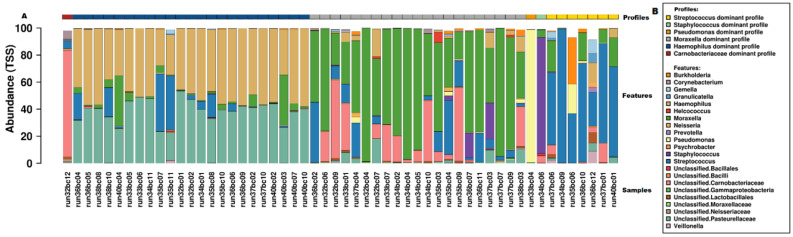
Microbiome profiles of samples presented as relative genus abundance. (**A**) Bar chart of individual samples, displaying the relative abundance on genus level for the top 20 genera. Individual samples are presented as run numbers and the corresponding microbiome profile is presented by colored bars at the top of the figure. (**B**) Legend for figure A.

**Table 1 microorganisms-09-01446-t001:** Baseline characteristics and management of the study population (n = 167).

General Characteristics	N or Median	% or IQR
Age in months *	17	9–34
Gender: male	104	62.3
Season of ED visit		
Spring	57	34.1
Summer	25	15
Autumn	41	24.6
Winter	44	26.3
**Clinical presentation**		
Duration of ED stay in minutes * (n = 141)	147	112–193
Temperature at ED visit	38.8	1.0
Duration of fever in days* (n = 166)	2	1-3
Tachypnea according to APLS (n = 165)	145	87.9
Tachycardia according to APLS	127	76
Saturation < 94% (n = 166)	39	23.5
Capillary refill time prolonged (n = 166)	17	10.2
Retractions (n = 166)	105	63.3
Ill appearance (n = 165)	70	42.4
**Diagnosis**		
Pneumonia	88	52.7
Bronchiolitis	22	13.2
Upper RTI	36	21.6
Subglottic laryngitis	2	1.2
Viral induced wheeze	18	10.8
Other	1	0.6
**Investigations**		
CRP in mg/L * (n = 152)	35.5	15–76
Other lab tests performed (n = 161)	41	25.5
PCR for viral diagnostics performed (n = 161)	45	28
Chest X-ray performed	35	21
**Management and outcome**		
Hospitalization at first visit	107	64.1
Hospitalized at any time	111	66.5
Duration of hospitalization in days (n = 164) *	1	0-3
Strategy failure (n = 161)	36	21.6
Antibiotics at first visit	85	50.9
Antibiotics at any time (n = 162)	100	61.7
Risk groups based on Feverkidstool ^a^ (n = 146)		
Low	30	20.5
Medium	33	22.6
High	83	56.8
Predicted risk of bacterial pneumonia by feverkidstool (n = 146) *	12.3	5.6–29.0
Oxygen therapy received at any time	87	52.1

Data are presented as total n and percentage or median with interquartile range indicated by *. “Strategy failure” was defined by a predefined composite outcome and was based on disease course during the 7 days after ED presentation. ^a^ Risk groups are based on the calculated predicted risk of having bacterial pneumonia using the Feverkidstool. If any data were missing, total n for that specific characteristic is reported. Definition of abbreviations: IQR = interquartile range, ED = emergency department, APLS = advanced pediatric life support, RTI = respiratory tract infection, CRP = C-reactive protein, PCR = polymerase chain reaction.

**Table 2 microorganisms-09-01446-t002:** Viral/mycoplasma infections and co-infections detected by PCR.

	N positive	ADV	boca	corona	entero	hMPV	infl A	infl B	Myco	parainfl	rhino	RSV
	N (%)	Count	Count	Count	Count	Count	Count	Count	Count	Count	Count	Count
Adenovirus	21 (13.3)	(2)										
Bocavirus	11 (7.0)	4	(1)									
Coronavirus	10 (6.3)	2	0	(2)								
Enterovirus	14 (8.9)	2	0	1	(6)							
hMPV	20 (12.7)	1	1	1	1	(15)						
Influenza A	5 (3.2)	0	0	0	0	0	(5)					
Influenza B	2 (1.3)	0	0	0	1	0	0	(1)				
Mycoplasma	2 (1.3)	1	0	0	0	0	0	0	(0)			
Parainfluenza	14 (8.9)	2	1	1	2	0	0	0	0	(8)		
Rhinovirus	63 (39.9)	12	6	4	6	2	0	1	1	5	(33)	
RSV	39 (24.7)	6	3	4	2	1	0	0	1	1	8	(21)

Data are presented as the number of pathogen co-infections. The values between () indicate the number of single virus infections for that specific pathogen. Definition of abbreviations: ADV = adenovirus, boca = bocavirus, corona = coronavirus, entero = enterovirus, hMPV = human metapneumovirus, inlf = influenza, Myco = Mycoplasma pneumoniae, parainfl = parainfluenza virus, rhino = rhinovirus, RSV = respiratory syncytial virus.

**Table 3 microorganisms-09-01446-t003:** Unadjusted and adjusted associations of viruses and microbiome profiles with hospitalization.

Pathogen	Unadjusted OR		95% CI	sig	Adjusted OR	95% CI	sig
Adenovirus	0.46		0.182–1.161	0.1	0.388	0.137–1.095	0.074
Bocavirus	2.69		0.561–12.907	0.216			
Enterovirus	0.076		0.016–0.353	0.001	0.078	0.015–0.395	0.002
Coronavirus	1.37		0.340–5.519	0.685			
Rhinovirus	1.087		0.559–2.114	0.805			
hMPV	1.055		0.395–2.818	0.915			
Influenza A	0.842		0.136–5.193	0.853			
Parainfluenza	0.532		0.177–1.602	0.262			
RSV	1.368		0.631–2.968	0.427			
**Virusgroups**							
High virulence	1.167		0.411–3.315	0.772			
Low virulence	0.899		0.310–2.607	0.845			
Negative	ref						
PCR positive for virus	1.038		0.384–2.807	0.941			
**Microbiome profiles**							
Haemophilus	ref				ref		
Moraxella	1.129		0.287–4.441	0.862	1.303	0.268–6.343	0.743
Streptococcus	2.118		0.210–21.389	0.525	2.877	0.220–37.641	0.420

Data are presented as odds ratios with 95% confidence intervals and *p*-value as calculated by logistic regression. In the multivariate models, odds ratios were adjusted for age, gender, detection, and virus in the nasopharyngeal sample and Feverkidstool risk groups. Definition of abbreviations: OR = odds ratio, CI = confidence interval, hMPV = human metapneumovirus RSV = respiratory syncytial virus.

## Data Availability

Individual participant data, virology data and microbiota data that underlie the reported results will be available after de-identification at time of article publication, ending 10 years following article publication. Data will be shared with investigators who provide a methodologically sound proposal, or for individual participant data meta-analysis. Data are deposited in the repository of Data Archiving and Networked Services (DANS, doi: 10.17026/dans-27a-fj4k). Proposals should be directed to info@dans.knaw.nl; to gain access, data requestors will need to sign a data access agreement.

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
