# Peer review of "The Role of the Respiratory Microbiome and Viral Presence in Lower Respiratory Tract Infection Severity in the First Five Years of Life"

_microorganisms, 2021, doi:10.3390/microorganisms9071446_

Round 1

Reviewer 1 Report

The affiliations should be checked, some departments are the same, and they are presented like everyone is different.

Introduction

Line 91: Remove “ small-scale study,”.

Material and Methods

Lines 105-106: The participating centers should be mentioned.

Lines 106-107: Develop a little more the exclusion criteria, at least the most important.

Line 121: Define “CRP point”.

Lines 129-131: It is unclear when the sample was taken and how many per individual?. It should be better presented.

Lines 132-135: The DNA extraction protocol and sequencing, although more extended in a different manuscript and the supplementary material, should be briefly described in this section.

Lines 136-138: Change “similar to several publications examining the nasal microbiome using Illumina sequencing” to as “it has been standardized in other nasal microbiomes studies based on Illumina sequencing.”

Lines 138-139: Remove “After exclusion of samples with insufficient reads and incomplete data,” since it is not necessary in this context.

Line 145: Capitalize the first m in Mycobacterium.

Lines 150-152: Both sentences should be rewritten to a better understanding of the procedures.

Lines 170-171: The richness is also considered an alpha-diversity metric. Include what metrics you have used as richness measure. Change “Richness and alpha diversity (Shannon index)” to richness (?) and diversity (Shannon).

Lines 174-178: Include the objective of these analyses. What have you used them for?.

Results

Lines 199-200: Table 1 contains too much information for the main text. I suggest preparing Table 1 with the most relevant clinical information for the study and including the rest of the data in a supplementary table.

Lines 234-235: I think you refer to p-value > 0.05 instead of 0.15. Please, clarify it.

Lines 243-244: It is unclear how the 6 clusters were defined visually from the figure or derived from some specific distance. Please, develop this. Also, it should be better connected with the NMDS analyses.

Lines 252: What is the meaning of “At sequence read level”?. Please, re-write this.

Discussion

Lines 386-388: You claimed that the analyses were performed at 10 000 reads depth although in the material and methods you indicated 1000. Please, correct it.

In limitations, the type of sample should be discussed. Some studies indicate that the respiratory tract microbiome is better represented by other samples such as tracheal aspirates, sputum, or BALs. The same analyses but based on other samples might cluster better microbiome, viruses, and severity profiles.

Reviewer 2 Report

The topic of the work is relevant for the clinical implications it may have for the prediction of potential severe diseases in young children who come to the hospitals.

In this study, the authors analyzed children aged one month to five years with fever and symptoms of respiratory tract infection who presented to the emergency room. The microbiome and the presence of viruses were studied in samples taken from the nasopharynx and results compared with rate of hospitalization and disease severity. Although some findings are not significantly associated with hospitalization or disease severity they are adequately explained and discussed. The study limitations are also well exposed.

There is only one error on page 9 figure 2: the color of the Carnobacteriaceae dominat profile is not the same in figure A and in the legend B.
